# Dairy Farmers and Veterinarians’ Agreement on Communication in Udder Health Consulting

**DOI:** 10.3390/vetsci11120665

**Published:** 2024-12-18

**Authors:** Michael Farre, Erik Rattenborg, Henk Hogeveen, Volker Krömker, Carsten Thure Kirkeby

**Affiliations:** 1Department of Veterinary and Animal Sciences, Section for Production, Nutrition and Health, University of Copenhagen, DK-1870 Frederiksberg C, Denmark; 2SEGES Innovation, Agro Food Park 15, 8200 Aarhus, Denmark; 3Business Economics Group, Department of Social Sciences, Wageningen University and Research, 6706 KN Wageningen, The Netherlands

**Keywords:** udder health, consulting, herd veterinarian, dairy farmer, communication

## Abstract

The herd veterinarian plays a crucial role in providing information and troubleshooting on udder health and bovine mastitis on a dairy farm. In this study, we questioned pairs of dairy farmers and herd veterinarians to uncover the dairy farmer’s perception of the communication with the herd veterinarian’s skills as a consultant. We found that agreement between the dairy farmer and the herd veterinarian in terms of interaction was surprisingly poor. On that basis, we suggest that the herd veterinarian should rethink their approach to motivating, implementing, and monitoring the quality of udder health consulting. The frequency with which udder health is discussed and how potential problems are identified were associated with a significantly lower BTSCC estimate of 427 cells/mL. In contrast, general cooperation between the dairy farmer and herd veterinarian was associated with an estimated increase of 604 cells/mL in BTSCC.

## 1. Introduction

The dairy sector faces many challenges due to societal pressure to handle the negative environmental effects of dairy production. A reduction in the number of dairy farms and increasing herd sizes have become global trends over recent decades [1], thus impacting the sector’s development. Increases in herd size are characterized by the farm’s transformation from a family labor-driven farm to the need for hired labor—often immigrants in the industrialized world [2]. In the period when the dairy farms’ structure changed, the dairy veterinarian tasks changed from individual animal treatment to proactive herd health management [3,4].

During this period, dairy veterinarian-initiated herd health services and preventive veterinary medicine shifted from the traditional approach of reactive ‘firefighter’ work and drug distribution [3]. This third phase was characterized by a shift from reactive to proactive involvement of the herd veterinarian, as dairy farmers recognized that increased herd performance was linked to monitoring and reducing the negative impact of subclinical disease. Dairy veterinarian-initiated herd health services and preventative veterinary medicine changed in that period from the historical approach of firefighter work and distributing drugs [5]. 

As a result, the dairy veterinarian of today needs a comprehensive soft-skills toolbox with a variety of skills that may be very different from the traditional veterinary college curriculum to have an impact on the bulk tank somatic cell count (BTSCC) as a measure of udder health [6,7]. The herd veterinarian must acknowledge and understand the dairy farmer’s aim—whether to eradicate disease, increase production, or achieve other goals [8]. The increase in the production level may also increase the incidence of production-related diseases (Gröhn et al. (1995)) [9], and dairy veterinarians today therefore need an economic mindset to allow them to focus on the costliest production diseases in dairy cows [10]. 

Dairy veterinarians must therefore be skilled in interpersonal communication at different operational levels and frequently in a foreign language to accommodate different owners and a diverse labor force. Evidence shows that the involvement and respect of experience in relations between patients or clients improve consultancy quality in human and veterinary medicine [11,12]. Also well described in the literature is the use of motivational interviews by Bard et al. (2018) and Svensson et al. (2020) [13,14], which can be another add-on approach to improving communication. Therefore, the herd veterinarian needs to adapt to what is regarded as necessary by the dairy farmer (Kristensen and Enevoldsen (2008)) [15] and balance the time each of the two parties participates in the mutual discussion during the general herd health consulting [16]. All these factors have been identified as essential to reaching the overall goal in herd health programs, where association with improved performance, e.g., milk production and BTSCC, was an expected outcome [17]. Researchers have focused on different ways to implement knowledge at dairy farms, mainly within udder health consulting, to facilitate this development. One approach has been the “stable school”, where farmer-to-farmer communication catalyzes the implementation of changes [18]. In terms of communication between dairy farmers and dairy veterinarians, previous studies have identified some challenges, with a client group that dairy veterinarians typically regard as “hard to reach” when they are focused on improving udder health on the farm [19]. However, most dairy farmers acknowledge the herd veterinarian is a crucial partner in udder health consulting [7]. Therefore, the successful herd veterinarian must identify herd-specific goals and targets and ensure that communication is tailored to the specific dairy farmer (Jansen et al. (2010)) [20], while considering the farmer’s aim and values. As recommended by Bard et al. (2028) [13], trust, shared understanding between the dairy farmer and herd veterinarian, and the meaningful interpretation of advice are critical factors for motivating change. Therefore, good consulting consists of several elements e.g., herd-specific goals, tailored communication, trust, and understanding, and to ensure this, we need to address the level of agreement between the dairy farmer and the herd veterinarian.

Nevertheless, to our knowledge, no previous studies have aimed to determine the level of agreement between dairy farmers and herd veterinarians in the mutual communication during udder health consulting and how this may affect BTSCC. The expected results can be utilized to target the future training and communication between the dairy farmer and the herd veterinarian, with the benefit of improved mutual communication between the stakeholders involved. 

Therefore, this study aims to determine the agreement in mutual communication between dairy farmers and herd veterinarians working with udder health consulting in Denmark, where the agreement in communication affects the BTSCC. Ideally, the dairy farmer and herd veterinarian relationship is built on mutual trust and an alignment of expectations; therefore, we hypothesize that the agreement would be at least minor to moderate, and the association with BTSCC will be minor. Specifically, the study aims to investigate two critical questions: What is the level of agreement between dairy farmers and herd veterinarians in terms of their perception of mutual communication when it comes to udder health? Does the level of agreement in mutual communication between the herd veterinarian and dairy farmer affect the BTSCC.

## 2. Materials and Methods

### 2.1. The Recruitment of Farmer-Veterinarian Pairs

This study evaluated herd veterinarians’ udder health consulting skills by asking dairy farmers and herd veterinarians similar questions regarding the consulting process. The following section details the paired dairy farmer and veterinarian recruitment process, the questionnaire development, and the statistical analyses. As the first step, we defined several essential enrolment criteria to target the predominant herd type in Denmark: milking in a parlor/rotary, >90% pure breed Holstein cows, no organic dairy herds, milk recording (DHI recording), herd size > 100 cows, and postal code > 6000. We then decided to focus the fieldwork on the western part of the country, where most dairy herds are located. We retrieved information from the National Danish Cattle Database. We identified a sample of 498 eligible herds that met the above-mentioned criteria. We randomized the eligible herds and imposed the inclusion criteria, resulting in (n = 88) herds included in the study, which we deemed a feasible number regarding our resources. We contacted the dairy farmers by telephone to encourage them to participate. If they agreed to participate, the researcher visited the dairy farms two times. All participating dairy herds had a herd health contract with a herd veterinarian, including mandatory herd visits 1–3 weeks apart, depending on herd size. This contract is required for herds with >100 milking cows, and we asked each dairy farmer to confirm that they had a herd health contract during the initial contact. The contract requires general monitoring of the herd’s health status, with particular attention given to areas with the highest consumption of antimicrobials. Thus, this study is set up to study if they agree about the way they communicate—not what the intervention is or what they decide to do when they have what they regard is a problem, which is highly subjective. The herd veterinarian had the contract for more than 12 months at the initial contact. The herd veterinarian would analyze and comment on the udder health status of the dairy farm at minimum every three months and suggest interventions where required according to the target. We informed the dairy farmer about the purpose and aim of the research project during the initial contact to ensure transparency and to ask for consent. 

We then contacted the corresponding herd veterinarian and provided the same information to motivate them to participate in the research project. All dairy farmers and herd veterinarians approved the data collection of the questionnaire data and the publishing of anonymous results from the research project. Data were then collected during an 18-month period, from August 2019 to February 2021. 

### 2.2. Questionnaire

We developed two questionnaires—one for the dairy farmers and one for the herd veterinarians—to identify the interaction between these two participants. Each dairy farmer and herd veterinarian filled in the paper questionnaire that included 40 questions for the dairy farmers Appendix A and 37 questions for the herd veterinarians Appendix A. The questionnaire was primarily based on the Likert scale (1932) [21], with the following options: strongly agree, very agree, agree, neither, disagree, very disagree, and strongly disagree. The questionnaires were divided into five groups of questions, with different topics. The two questionnaires were similar, though some questions were phrased slightly differently according to whether they were targeted toward the dairy farmer or the herd veterinarian. These questions were descriptive and were therefore not included in any further analysis. One question could be if there was a social interaction as part of the time invoiced to the farmer.

The answers to each group of questions were addressed in the further analysis as construct pooling questions selected to represent (1) how the dairy farmer was involved in the discussion on udder health (C1), (2) the frequency of follow-up and identification of potential problems (C2), (3) the implementation of SMART action plans (C3), (4) general cooperation between the dairy farmer and herd veterinarian (C4), and (5) attitudes toward udder health consulting, animal welfare, antimicrobial consumption, and consumer perception (C5). We used Excel^®^ sheets with drop-down menus to avoid the entry of nonsensical answers in recording questionnaire data and to restrict the number of errors. 

### 2.3. Data Collection

Each dairy farmer was visited twice; during the first visit, the dairy farmer filled in the questionnaire and the first author could briefly clarify any question that was unclear as needed. If they had concerns regarding the questions, a third party read them, and the dairy farmer answered. The second visit was a follow-up visit where results from another part of the project not related to the questionnaire was discussed. At the same time, the questionnaire was sent to the herd veterinarian via email, and they then returned the questionnaire to the researcher in person or via email. Herd veterinarians who did not return a completed questionnaire within two weeks of the farm visit received a reminder with an invitation to participate. The first author handled all the data management of the questionnaires to ensure consistency and contacted the participating dairy farmers and herd veterinarians. The first author also retrieved information from the National Danish Cattle Database regarding herd BTSCC.

All data management and analyses were conducted using the statistical software R (Version 4.2.0, R Core Team 2022, https://www.R-project.org/, accessed on 15 June 2022) The first analysis aimed to establish the level of agreement for each construct regarding the perception of communication between the dairy farmer and the herd veterinarian during udder health consulting. The agreement was calculated using Cohen’s weighted kappa (κ) applied to ordinal-scaled variables, which takes a systematic difference in the level of the answers into account, as described by Houe et al. (2004) [22]. The author then matched the answers from the dairy farmer and herd veterinarian if they were on an ordinal scale (n = 25 questions). If the questionnaire from the herd veterinarian was missing, the data from the dairy farmer were excluded. We used the following guidelines to interpret κ: less than 0.4 indicated poor agreement, 0.4–0.75 indicated fair to good agreement, while over 0.75 indicated excellent agreement [23]. The second analysis employed a linear regression model to identify factors significantly associated with BTSCC. To do this, five constructs of questions were created to address potential communication issues related to BTSCC. The questions were assigned to each of the five constructs based on expert assessment by the author. Each construct included five to eight questions that supported the overall classification of the construct. Within each construct, the agreement between the dairy farmer and herd veterinarian answers was calculated for each farm and used as an explanatory variable for the farm.

Herd size was included in the model as a discrete variable and was considered a confounder, affecting both BTSCC and agreement, as illustrated in the DAG diagram in Figure 1. 

The BTSCC was regarded as normally distributed, and the BTSCC data were from the past 12 months before the visit. We then constructed a linear regression model to identify factors affecting the dependent variable BTSCC: BTSCC~Agreement construct + herd size

The questions for the dairy farmer and herd veterinarian were transformed into the defined constructs, and the allocation of the questions to each construct was carried out before the data collection began and was based on the experience of the authors. They allocated the questions based on the headline for the construct and added the question where they agreed it was truthful. The last herd size was included in the model as an explanatory variable. We then used a backward elimination procedure, and the model was designed to identify constructs that significantly affected the outcome, with *p* ≤ 0.05. Missing values were excluded from the analysis.

## 3. Results

The descriptive statistics were first calculated; there were four herd veterinarians answering the questionnaire on request, and one was incomplete because the herd veterinarian had never discussed udder health at the farm. Finally, one herd veterinarian had herd health contracts with two different dairy farms enrolled in the study. We enrolled dairy farmers and herd veterinarians from the same herds, and the response levels were high, with 100% of dairy farmers participating at the time of our visit, but the herd veterinarians also had a high questionnaire return rate of 94%. The first author verified the answers but did not address potential missing values at the visit. No illogical answers were recorded because the categories for answers were fixed. 

The herds enrolled ranged from 105 to 1291 cows, with a mean herd size of 326 milking cows. There were three milking years per culled cow, calculated from first calving to leaving the herd as a culled cow, and a rolling average of 11,774 L of energy-corrected milk. The mean, median, and quartiles of the farm-level geometric mean BTSCC were then calculated, with the results shown in Table 1. 

The minimum and maximum agreement values of each of the five constructs are illustrated in Figure 2.

The herd veterinarians visited the farms every 1–3 weeks according to legislation and depending on the herd size. The perception data included are from questionnaires collected from (n = 88) dairy farmers and (n = 84) veterinarians.

The complete output from testing Cohen’s weighted kappa value for paired questions can be found in the Appendix A. As a summary, the agreement between dairy farmers and herd veterinarians regarding communication for each paired question is graphically illustrated in Figure 3. The agreement ranged between −0.06 and 0.12, indicating no to slight agreement, which was lower than the expected moderate agreement. 

This suggests that the dairy farmers and herd veterinarians perceived the communication of the udder health consultation quite differently. 

There were a few differences between farmer and veterinarian answers in the constructs, as illustrated in Figure 2. From the dairy farmers’ perspective, in question pair two, representing “To what extent do you agree that there is an opportunity for fruitful discussion between the farmer and herd veterinarian, with time for reflection during the meeting?” (Appendix A), the agreement was −0.06. With an agreement at 0.12, question pair 22, “To what extent do you agree that mastitis is a serious problem in the herd, bearing in mind the sector’s target of reducing antibiotic consumption” (Appendix A), had the highest agreement in the study. 

An important result was the factors associated with udder health, represented by the BTSCC, here in Table 2, the full and the final model can be seen. After the calculations, constructs, C1, C3, and C5 (Appendix A) were eliminated, as well as herd size. 

The coefficient for construct (2), the frequency with which udder health is discussed and how potential problems are identified, was negative; every unit agreement increase was associated with a 427 cells/mL lower BTSCC. In contrast, for construct (4), general cooperation between the dairy farmer and herd veterinarian, in investigating the understanding and relationship between them, the coefficient was positive but small; one-unit higher agreement between the dairy farmer and herd veterinarian was associated with a 604 cells/mL higher BTSCC. 

## 4. Discussion

To continuously improve communication between dairy farmers and their herd veterinarians, we first need to obtain the present status. Therefore, we assessed the agreement between the dairy farmers and herd veterinarians’ perception of the mutual communication during udder health consultation and found no to slight agreement (Fleiss (1981)) [23], which was between −0.06 and 0.12. This finding was in contrast with our expectations that the agreement would be at least moderate to have a working relationship between the dairy farmer and herd veterinarian. Good consulting from the herd veterinarian consists of several elements, e.g., herd-specific goals, tailored communication, trust, and understanding; therefore, we analyzed the level of agreement. In other areas of veterinary medicine, successful communication has been linked to quality (Abood (2008), Churchill and Ward (2016)) [24,25]; therefore, we associated the agreement in perceived communication with the actual BTSCC on a farm. 

In conducting this type of research, it is always a challenge to prevent self-reporting bias in a questionnaire, where the respondents potentially rate themselves higher, especially in areas with adverse effects [26,27]. This issue must be considered when assessing the agreement. A second challenge in self-reporting questionnaires is the impact of negative mood states in the past two weeks prior to answering the questionnaire [28]. This potential bias in answering must be considered because we focus on asking about the most recent visit from the herd veterinarian to avoid the risk of recall bias. Thus, we wanted to focus on the degree to which they agreed and whether certain areas would indicate agreement to monitor the status and highlight whether and potentially where improvement was needed. There can also be some contextual differences between the two groups of respondents, which impact the distribution of answers and, therefore, the level of agreement. This point can be affected by the different educational backgrounds of dairy farmers and herd veterinarians, social norms and values, and social influence from other peer groups. The respondents’ personalities also impact the responses, where some will choose extreme responses such as “strongly agree” or “strongly disagree”, potentially leading to low agreement. Also, here, the dairy farmer has to relate only to the herd veterinarian and only one person; the herd veterinarian is also asked about the specific dairy farmer. However, the herd veterinarian has a group of farmers they service; therefore, they could risk providing answers of more general opinion.

Construct 2 was found to have a confidence interval of the estimate overlapping with zero. This indicates that this variable could be insignificant. Still, it was kept in the final model by the backwards model selection procedure using AIC. Further studies should address this because a small but consistent effect might have a CI that overlaps with zero if the effect size is too small to be reliably distinguished from zero due to high variability or limited precision.

The question pair with the least agreement was question pair 2, which was a personal question regarding satisfaction: “To what extent do you agree that there is an opportunity for fruitful discussion between the dairy farmer and herd veterinarian, with time for reflection during the meeting?”. Because we have this low level of agreement, we need to reflect on the extreme values in trying to understand the challenges in communication between the dairy farmer and the herd veterinarian. It seems like we need to work with the frame of the meeting—because the discussion lacks mutual understanding. The meeting is set, and the topic discussed is the udder health, but they do not discuss it out of a common understanding of the aims and goals of the dairy farmer.

In contrast, question pair 22 had the highest agreement, where the question was more general regarding antimicrobials: “To what extent do you agree that clinical mastitis in cows requires excessive resources and time from the farm employees?” (Appendix A). The latter question, which received the highest level of agreement, is more technical in nature and likely easier for participants to agree on. In this context, both dairy farmers and herd veterinarians are less influenced by personal feelings, preferences, or individual personalities regarding the goals and objectives for the dairy herd. Instead, their shared practical experience in managing sick cows allows them to readily understand the complexities of diagnosing, treating, and handling waste milk, which is often viewed as an inconvenient disruption to the daily routine. And this point about the disruption is important, because continuity is very important in milk production to ensure the allocation of enough resources in terms of employees at any given time.

Handling antimicrobials was described by Gröndal et al. (2023) [29], where prescribing antimicrobials for dairy cattle needs to be understood as taking form in a relationship in which both veterinarians and farmers take an active part. This can be a reason for the level of agreement. 

The scope of the seven-point Likert scale challenged the likelihood that the dairy farmer and herd veterinarian have the exact same answer to their perception of communication in udder health consulting at a specific dairy farm, but this was overcome with the weight of the answers. Other researchers found that the herd veterinarian’s perceived skills were not equal to those desired by the dairy farmer [30]. Here, the herd veterinarian could implement the motivational interview to improve agreement, which would be beneficial to improving the overall communication during consulting [31]. However, the perspectives of the dairy farmer and the dairy veterinarian still differ because their starting point for conversation differs. We did not expect complete agreement, but a discrepancy of this magnitude was unexpected. 

The multivariable analysis for associating BTSCC with constructs representing agreement on communication in udder health consulting showed that monitoring DHI data related to udder health and regular follow-up by the herd veterinarian in construct (2) “The frequency with which udder health is discussed and how potential problems are identified” were associated with a significantly lower BTSCC of 427 cells/mL. This agreement was expected because the role of monitoring has long been seen as necessary: “Udder health monitoring is an essential part of preventive veterinary medicine [6]. Another significant factor associated with BTSCC is the agreement between dairy farmers and herd veterinarians in construct (4), general cooperation between the dairy farmer and herd veterinarian, in investigating the understanding and relationship between them; more cooperation was associated with a 604 cells/mL higher BTSCC. 

This result is, on one hand, counterintuitive but like the findings of Stevens et al. (2019) [32], who found increased new infection rates in the herds with increased veterinary intervention. Conversely, dairy farmers with more udder health problems may cooperate more to solve these issues. Therefore, we need to consider plausible explanations to interpret the findings of increased cooperation and high BTSCC, such as (1) a recent acute clinical mastitis problem encouraged the farmers to seek advice from the herd veterinarian, (2) the dairy farmers are in the process of improving the BTSCC and therefore feel confident and recognize the value created by the herd veterinarian, or (3) the dairy farmer or the herd veterinarian over- or underestimate the advice. It is evident from the present analysis that the herd veterinarians may not be fully aware of the farmers’ aims and targets. This observation agrees with Derks et al. (2012) [33], who found that dairy veterinarians providing services in herd health programs did not sufficiently align with the dairy farmer regarding goal-setting and evaluation. This makes it difficult to solve potential herd-specific problems and add value to the dairy farm. It is, of course, easy to point at the herd veterinarian, who must be proactive in providing the service. Yet, it is also essential to consider how skilled the farmer is in defining their own needs and be instrumental in implementing the herd-specific advice from the herd veterinarian. 

Thus, from a business perspective, the dairy farmer should select a herd veterinarian who is skilled and capable of suggesting interventions to solve problems, focus on communication (Derks et al. (2012)) [33], and use communication methods like motivational interviewing for improved implementation [34]. This points to self-reflection for the herd veterinarians who want to be the preferred dairy consultant in the future and adapt their communication style towards the dairy farmer’s needs [31]. Herd veterinarians will face more challenges due to the development of larger farms, more hired staff, and several management levels. As a result, herd health veterinarians must act as integrated team members and be able to adapt their communication style, motivation, and operational effort to each dairy farmer they work with. In that context, other researchers have suggested four categories of dairy farmers (Jansen et al. (2010)) [19]: proactivists, do-it-yourselfers, wait-and-see-ers, and reclusive traditionalists, but we still need more insight into how we can successfully approach each type. The herd veterinarian needs to consider the different kinds of dairy farmers to target how they approach their problems. Effective communication will only become more important because even though automation and the increased application of milking robots and sensor technology will remove the need to align with some operational staff, clear communication and establishing specific and realistic goals with the dairy farmer is still essential. The dairy farmer expects value for money when involving the herd veterinarian, and the herd veterinarian must continuously offer herd-specific solutions to add value to the dairy farm. However, farmers differ in their opinions and behavior, and we therefore expected variation in the “Cooperation with the herd veterinarian” construct regarding the level of agreement. The five constructs also differ in importance. 

Of the constructs with no significant outcome, it was interesting that a SMART action plan was not associated with better udder health. Thus, the herd veterinarians should try to identify the dairy farmers’ clear and well-considered vision for improving udder health consultation and deal with dairy farmers individually. The well-informed dairy farmers only need support from the sideline with motivation and knowledge upon request. In contrast, the herd veterinarian must suggest goals and actions for the less informed farmers and provide more operational support. Researchers such as Kupier et al. (2005) [7] found that the herd veterinarian is an essential partner for the dairy farmer, and we anticipated that the dairy farmer and herd veterinarian would regularly discuss and align expectations regarding udder health consulting. 

However, our analysis demonstrated a wide variation in opinion about the communication between farmers and their veterinarians during udder health consulting. The current study included (n = 88) dairy farmers and their consulting herd veterinarians, compromising the study’s statistical power. Other studies, such as Jansen et al. (2010) [19] had (n = 25) dairy farmers enrolled: Falkenberg et al. (2019) [35], with (n = 33) dairy participating veterinarians, and Bard (2018) [13], where (n = 14) dairy farmers and their corresponding herd veterinarians were interviewed. However, there is a difference in how the data were collected in the cited studies; they were based on interviews, whereas the present study is based on closed-end questions in a questionnaire. The studies based on semi-structured open-ended questions are beneficial for a more explorative approach, where other topics than the data from the present study can be retrieved. 

Another challenge of this study could be the choice of Likert scale steps in monitoring agreement; yet, research has shown that questionnaires become less accurate when the scales include less than four and more than seven points [36]. We therefore chose Likert scale questions with answers in five to seven steps in accordance with other studies on mastitis and udder health consulting [32]. The scale is sensitive to a single-unit difference in answer, but weighting was used to compensate for this. The high level of response allowed us to analyze the impact of perception at the herd level and helped reveal a low level of agreement. 

Based on our analysis, it is evident that there is a need to rethink the collaboration between dairy farmers and herd veterinarians in the operationalization of udder health management at the herd level. From the perspective of the herd veterinarian, there is significant potential for more active engagement with communication strategies that have been empirically shown to be effective, such as motivational interviewing [31]. This evidence-based technique emphasizes empathetic listening, active engagement, and maintaining a non-judgmental stance. Motivational interviewing fosters a collaborative dynamic where the focus is on resolving ambivalence and enhancing the individual’s intrinsic motivation to embrace change [37].

However, the successful implementation of this approach hinges on the clarity of the dairy farmer’s objectives and the establishment of interventions framed within a SMART action plan [38]. By aligning goals and interventions in a well-defined, structured manner, the SMART approach not only ensures that the aim is clear but also facilitates the allocation of shared responsibilities between the farmer and the herd veterinarian. This communication strategy represents a positive shift toward more effective collaboration, where the veterinarian’s guidance is paired with a clear roadmap that drives accountability and tangible outcomes.

In the current study, several limitations must be considered when interpreting the results. First, there may be over- or under-reporting, as some respondents might feel a social responsibility to provide confident answers, particularly in the context of the customer–service provider relationship. Additionally, some respondents may prefer extreme statements on the Likert scale, which could skew the results. Lastly, selection bias based on the inclusion criteria may also be present—for example, dairy farmers using robotic milking systems might have different perspectives than those who do not.

A follow-up study could address some of these issues by incorporating triangulation, such as through personal interviews or focus groups, to add additional data and perspectives. Observational studies, which rely less on subjective self-reports, offer an alternative method for data collection and could help reduce many of the biases associated with self-reporting, as seen in the current study.

All through the limitations, this study fills a gap in the current literature and has contributed new knowledge to help improve the quality of udder health consulting. 

## 5. Conclusions

In a study on (n = 84) pairs of dairy farmers and their consulting herd veterinarians, we found an agreement between −0.06 and 0.12 in the perception of communication between farmers and veterinarians. Since communication between farmers and veterinarians during consulting is essential to deliver effective advice, it is therefore vital to further develop consulting skills, improve communication during the udder health consulting provided, as well as to align expectations, goals, and targets. We also found some association between perceptions of communication during udder health consulting and a reduction in BTSCC. The frequency with which udder health is discussed and how potential problems are identified were associated with a lower BTSCC estimate of 427 cells/mL. In contrast, in general, cooperation between the dairy farmer and herd veterinarian was associated with an estimated increase of 604 cells/mL in BTSCC. This study could be improved by increasing the sample size and include open-ended questions to motivate the respondents to elaborate on the topics they found most important.

## Figures and Tables

**Figure 1 vetsci-11-00665-f001:**
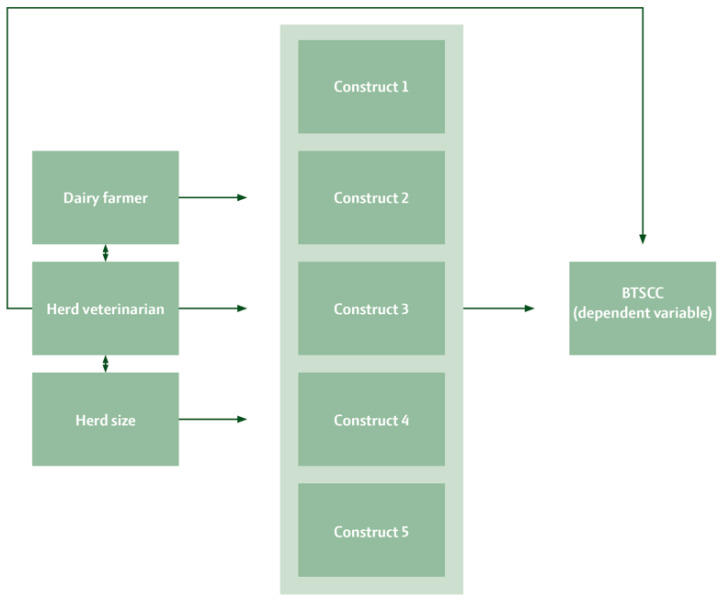
DAG diagram illustrating the association between the herd explanatory variables (**left**), each of the constructs used as explanatory variables (**middle**), and the dependent variable (**right**).

**Figure 2 vetsci-11-00665-f002:**
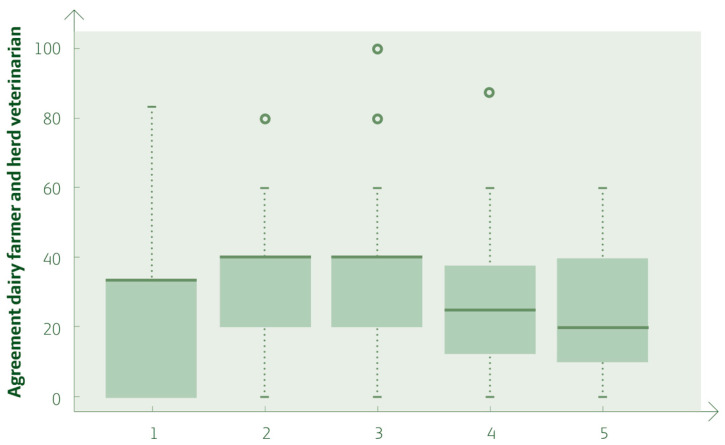
Boxplot showing the agreement between the farmers and the herd veterinarians for each of the five constructs in the study, with dots illustrating outliners.

**Figure 3 vetsci-11-00665-f003:**
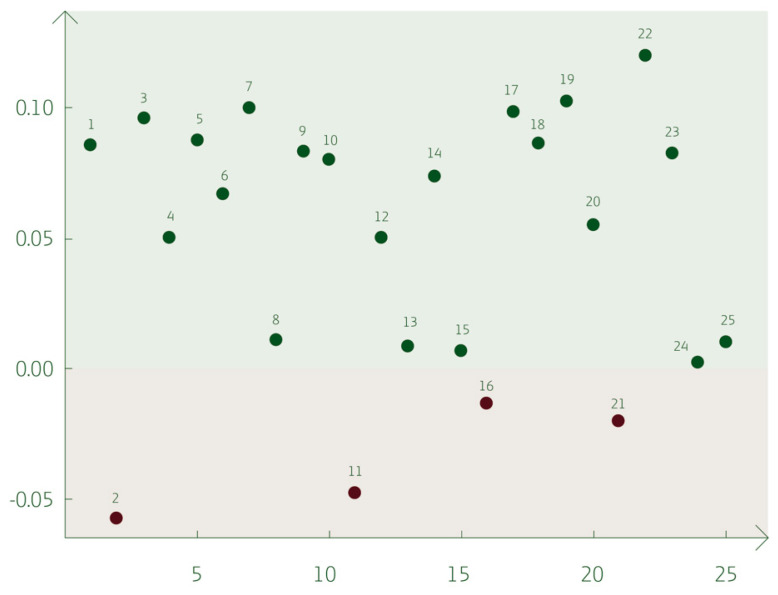
The agreement of the paired questions of the dairy farmer and herd veterinarian based on the Cohen’s weighted kappa value. The x-axis refers to the question number, and the y-axis refers to the Cohen’s weighted kappa value see Appendix A.

**Table 1 vetsci-11-00665-t001:** The descriptive statistics of BTSCC from the enrolled dairy herds in the perception study.

	Mean	Median	Min.	Q1	Q2	Q3	Max.
BTSCC cells/mL	207.000	204.000	89.00	177.00	204.00	237.00	362.000

**Table 2 vetsci-11-00665-t002:** Results from the full linear regression model, and the final model after backwards selection by AIC, with the coefficients, parameter estimates, 95% CI, and *p*-value.

			Full Model			Final Model		
Coefficients	Parameter Estimates	SEM	*p*-Value	95% CI	Parameter Estimates	SEM	*p*-Value	95% CI
Intercept	205.754	21.070	0.44		206.572	15.258		
Construct 1	−257	331	0.29	[−906; 392]				
Construct 2 ^1^	−349	324	0.71	[−983; 284]	−427	303	0.16	[−1028; 170]
Construct 3	−104	275	0.04	[−644; 435]				
Construct 4 ^1^	705	339	0.25	[41; 1370]	604	299	0.04	[13; 1.194]
Construct 5	349	301	0.71	[−242; 940]				

^1^ Final model coefficient.

## Data Availability

The datasets presented in this article are not readily available because the data are part of an ongoing study, which will be published soon. Requests to access the datasets should be directed to the corresponding author.

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
