# Peer review of "Dairy Farmers and Veterinarians’ Agreement on Communication in Udder Health Consulting"

_vetsci, 2024, doi:10.3390/vetsci11120665_

Round 1
Reviewer 1 Report
Comments and Suggestions for Authors
This study is well presented and shows a good level of detail of the methods used to make findings. Here are some comments and recommendations:
1. Line 24, spelling error, loos should be loss.
2. Line 153 (reference to Table 1 and Table 2 in Supplementary materials). The supplementary materials were not provided with this manuscript, so I am unable to see the included questions. It would be very helpful for the reader if the included questions could be made available to the reader. Perhaps my fifth comment below could be a way of addressing this.
3. Line 217. "Average" can mean different things. It is more accurate to refer to the mean herd size.
4. Line 217. When you say the average/mean herd was three milking years, do you mean 3 calendar years from first calving to cull date, or do you mean three lactations? I know these will be similar in herds with good reproductive performance, but they aren't exactly the same.
5. Results described in Figure 3. In the text you give question pair 2 and question pair 22, which is great because it allows the reader to know which questions were associated with the highest agreement and lowest agreement. Unfortunately the reader has no way of knowing what the other 23 question pairs are. For example I can see from figure 3 that pair 19 also has a high level of agreement, but I have no idea what the question or topic here actually is. It would be helpful for the reader if you could include all question pairs and their level of agreement (perhaps as a table). It would allow the reader to learn so much more about your findings. It would also be useful to know what question pair 11 is, as the level of disagreement between farmer and vet in question 11 was similar to the level of disagreement in question pair 2.
6. Lines 245-252 vs Lines 305-310. It is not clear why questions 2 and 22, when stated at lines 245-252 are not the same questions when stated again at lines 305-310. This may require more clarification.
Reviewer 2 Report
Comments and Suggestions for Authors
The manuscript, “Dairy farmers and veterinarians' perception of communication in udder health consulting,” aimed to describe the agreement in their perception of their communication during udder health consulting. From August 2019 to February 2021, 88 herds participated in the study, with herd size ranging from 105 to 1,291 milking cows. The dairy farmer and the herd veterinarian were asked to fill out a questionnaire to provide insight into their communication experience during herd health consulting. The agreement between farmers and veterinarians ranged from 0.06 to 0.12, indicating that the veterinarians' priorities for the herd are not aligned with the preferences of the dairy farmers.
The article was not well prepared. There are many mistakes in words and sentences, as well as redundant phrases and sentences. Please check the whole manuscript.
This article was constructed to be difficult to read. For example, there are rarely any paragraph breaks in the introduction and discussion parts, and all figures and tables are not self-understandable. Repeated sentences and phrases were found several times in the manuscript.
The introduction is quite long, with many redundant words and less specific information about the article's topic. In addition, the structure of the introduction is not in good order, making reading confusing. I would suggest revising it into 2-3 paragraphs for less than a page in the introduction part.
The description of the materials and methods in every paragraph is too long, redundant, and confusing. Please revise it to be shorter and clearer.
The description of the materials and methods used in this study is not clear. The authors described the target population as including farmers of 88 dairy farms, selecting based on the described criteria, matching with 84 veterinarians who responded to the herd health management program of the paired farm. The study was conducted for 18 months, from August 2019 to February 2021 (78 weeks). The farms were visited every 1 to 3 weeks according to the herd size (L116-L118). This means that each farm was visited and had the fill-out questionnaires from 26 times (for every 3 weeks) to 78 times (for every week). However, L121-124 indicated that the herd veterinarian would analyze and udder health status every 3 months, indicating that each farm would fill out the questionnaire 6 times. Please clarify the number of questionnaires used in this study.
For this kind of study, information on farms, farmers (n=88), and veterinarians (n=84) is needed.
Please describe in detail the five to seven answer options on an ordinal scale based on the Likert scale (1932).
Most questions in the questionnaires are about the levels of agreement (24 of 40). As the levels of agreement between 2 persons are generally not equal (especially when the levels are 5 to 7 levels, reducing the levels of agreement, for example, agree or not agree, would be better to find the agreement of 2 persons. I would suggest re-coding and re-analyzing.
Please explain: “The predefined weights measure the degree of disagreement; the higher the disagreement, the higher the weight.”
Many questions are related to the experience of a person, and the problem relates to a person. Please explain the method to minimize this problem.
Delete Figure 1.
Please revise the result part as it is quite confusing and complicated.
Please revise all legends of all figures and tables to allow them to have self-explanation.
In general, a person's level of agreement is based on their knowledge and concerns. At the time, the farm had high BMSCC, so it was more concerned about the BMSCC problem. How did the authors cope with this repeated data several times?
Please clarify the methods of re-coding the data from questionnaires for the factors used to evaluate their association with BMSCC.
The agreement construct was repeatedly explained several times. Please use the first one and delete the rest. It might be used as an abbreviation for better recognition.
The description of dependent variables, independent variables, and statistical analysis was not clear.
Please indicate the number of agreement and not-agreement for all questions.
Please check and revise Table 2 as some 95%CL might be wrong. In addition, please add information on SEM, statistical tests, and p-values to the tables.
Also, the discussion was not well-organized and was too long. Please revise it with a separate paragraph for each important finding.
Comments on the Quality of English LanguagePlease check English.
Reviewer 3 Report
Comments and Suggestions for Authors
This paper is excellent and clearly shows that veterinarians need to develop better methods for communicating with farmers. The sample was described clearly and the participation rate was high.
Line 18 - Add to the simple summary your results on somatic cell count. You found that high somatic cell counts were associated with poor communication.
Line 35 - Add to the abstract your results on somatic cell count.
Lines 107-120 - Really good complete description of the samples.
Reviewer 4 Report
Comments and Suggestions for Authors
The manuscript entitled "Dairy farmers and Veterinarians' perception of communication in udder health consulting" has been kindly reviewed. This study aimed to describe the agreement in perception of their communication during udder health consulting. Despite possible intetrests in this field, the current data is hardly to support the conclusions. Therefore, I would like to suggest authors to enrich the data, e.g., expand the effective sample size and increase the coverage of the questionnaire. Below are my specific comments:
(1) Title: The title does not adequately reflect the content and conclusions of this paper; it is suggested to make appropriate additions and modifications.
(2) The abstract is overly verbose in describing the background and provides scant and incomplete information about the results. For example, the last sentence only mentions that the data was analyzed without presenting the findings.
(3) Introduction: The introduction section lacks clear logic, and the cited references overlook the latest research developments, merely presenting the problem without proposing expected results. It is recommended to elaborate in separate paragraphs and add a description of the hypothesis section.
(4) Result: The data dispersion in the results section is too great, such as in Figures 2 and 3, which reduces the reliability of the data. This also indicates that the authors should increase the effective sample size.
(5) Discussion: The logic in the discussion section is also unclear; it can be elaborated in separate paragraphs corresponding to the results.
(6) Conclusion: The conclusion section should clearly state the results and conclusions of this study, and add the shortcomings of the research and directions for improvement, with language as concise as possible.
(7) Reference: Please delete the description: available from... and add more up to data references to enrich the contents here.
Round 2
Reviewer 2 Report
Comments and Suggestions for Authors
After carefully reading the revised manuscript, I see that it has not been improved. No significant revision was done based on the comments. No necessary information has been added. The manuscript remained confused and does not show any new knowledge from this study. In addition, some explanations make sure that the result of this study might be the problem of recall bias as the agreement was made between 1 year averaged BMSCC and the recent consultant. The researchers collected the farmer data, but the veterinarian data were collected by email. The agreement level was concluded from visits of the researcher only two times, meaning that all samples are 176 samples. Constructs 1-5 in Table 2 are not understandable in the detail of the factors inside. Again, the study is difficult to understand, redundant, and does not provide the necessary new knowledge. All findings of this study can be found in the textbook.
Author Response
"Please see the attachment."

Reviewer 4 Report
Comments and Suggestions for Authors
The author has made much improvements to the manuscript. I only have one other suggestion: enhance the clarity of Figures 1 and 3, as the current versions appear quite blurry.
Author Response
"Please see the attachment."

Round 3
Reviewer 2 Report
Comments and Suggestions for Authors
After carefully reading the latest revised manuscript, I still see that it has not been improved based on both previous comments. No necessary information has been added. I will try to comment in detail on each part.
The first paragraph of the introduction discussed the change in farm structure and introduced the proactive herd health program. The second paragraph explained more about the proactive herd health program. Then, veterinarians in the herd health program must have a variety of skills and acknowledge and understand farmers. Then, the veterinarians had to improve their communication skills to achieve better farm performance. The authors also stated that many improvements were implemented in communication. Some advice and problems of communication skills were identified in the introduction.
The pain point of this study on this study (L107-108) is that “no previous studies have aimed to determine the level of agreement between dairy farmers and herd veterinarians” might not be necessary because this level of un-agreement has been set up. So, we do not need any more research on this and can target future training and communication from now.
This makes this article low in originality/novelty. The introduction did not provide sufficient background on the essential reasons why we needed the level of agreement. Then, I would score on “not applicable”.
For designing the study, the information on the sample herds was retrieved from the National Danish Cattle Database. The authors selected the herd based on the criteria to target the predominant herd type in Denmark: milking in a parlor/rotary, > 90% pure breed Holstein cows, no organic dairy herds, milk recording (DHI recording), herd size > 100 cows, postal code > 6000. From 498 eligible herds, the author randomly selected only 88 farms (17.6%) based on the researchers’ resources without any further inclusive criteria and methods, for example, convenience and systematic randomization. In addition, no sample size calculation was performed for this selection.
All selected farms had a herd health contract with a herd veterinarian, including mandatory herd visits 1–3 weeks apart, depending on herd size. This indicated that each farm had a different management program because the contract every 3 weeks had less communication than every week for 3 times. The authors need to control this factor.
The authors said a farm contract with a veterinarian for at least 12 months was required for herds with > 100 milking cows. The contract requires general monitoring of the herd's health status, with particular attention given to areas with the highest consumption of antimicrobials. The sentence “The herd veterinarian would analyze and comment on the udder health status of the dairy farm at minimum every three months and suggest interventions where required according to the target” indicated that the udder health status would be performed at least every three months. My question is, “Why does it perform when the farm has a problem?”.
What kind of data is in the sentence (L157), “Data was then collected during an 18-month period, from August 2019 to February 2021”? It makes me confused about whether the data is research data or farm performance data. For the research data, the author collected only 2 times.
The two questionnaires with slight differences were developed for farmers and veterinarians, respectively. The answers to the questions were primarily based on the Likert scale (1932) [21], with the following options: strongly agree, very agree, agree, neither or disagree, very disagree, strongly disagree. As I said, the first comment is that the use of these answers is prone to be answered at different levels by farmers and veterinarians because both have different backgrounds in detail. So, I am not surprised by their conclusion about “the disagreement between veterinarians and farmers in this study.”
I asked for the authors the first time for this and did some statistical analysis that included the backgrounds of farmers and veterinarians, but I did not receive a response from them.
(L175) The answers to each group of questions were addressed in the further analysis as constructs pooling questions selected to represent 1) identifying how the dairy farmer was involved in the discussion on udder health (C1), 2) the frequency of follow-up and identification of potential problems (C2), 3) implementation of SMART action plans (C3), 4) general cooperation between the dairy farmer and herd veterinarian (C4), 5) attitudes toward udder health consulting, animal welfare, antimicrobial consumption, and consumer perception (C5).
Again, there is no explanation for pooling data together, and the methods of pooling were not clear. It will become only one score for each construct or something else.
L186, the authors explained the method of data collection again. They said, “Each dairy farmer was visited, during which the farmer filled in the questionnaire, and the first author could briefly clarify any question that was unclear as needed.” There seemed to be several visits, but there were only two in total.
L190, The sentence “At the same time, the questionnaire was sent to the herd veterinarian via email” means that the veterinarian answered the questionnaire by email. In addition, if there was no answer within 2 weeks, the veterinarian got a reminder. The authors did not indicate how much did the veterinarians get the reminder. Anyway, this duration could get the recall bias from the veterinarians.
L204. For data analysis, the author did not show a way of matching the answers of the dairy farmer and herd veterinarian. Did the score have to be the same for both? If so, it might be expected that the disagreement should be in this case.
For the second analysis, a linear regression model was used to identify factors (C1-C5) significantly associated with BTSCC. The BTSCC was from the past 12 months before the visit. The author said this is the average 12-month BTSCC. The questionnaire (C1-C5) was done during the visit, but the BTSCC was the average 12 months. This might not reflect each other. In addition, the individual BTSCC data generally had a non-normal distribution. Therefore, this analysis and the design of the study are not reliable.
This is all a serious design study. This means the results of this study are not reliable.
Author Response
"Please see the attachment."
